# Shallow Updates for Deep Reinforcement Learning

**Nir Levine**[*]
Dept. of Electrical Engineering
The Technion - Israel Institute of Technology
Israel, Haifa 3200003
`levin.nir1@gmail.com`

**Tom Zahavy**[*]
Dept. of Electrical Engineering
The Technion - Israel Institute of Technology
Israel, Haifa 3200003
`tomzahavy@campus.technion.ac.il`

**Daniel J. Mankowitz**
Dept. of Electrical Engineering
The Technion - Israel Institute of Technology
Israel, Haifa 3200003
`danielm@tx.technion.ac.il`

**Aviv Tamar**
Dept. of Electrical Engineering and
Computer Sciences, UC Berkeley
Berkeley, CA 94720
`avivt@berkeley.edu`

**Shie Mannor**
Dept. of Electrical Engineering
The Technion - Israel Institute of Technology
Israel, Haifa 3200003
`shie@ee.technion.ac.il`

**\* Joint first authors. Ordered alphabetically by first name.**

## Abstract

Deep reinforcement learning (DRL) methods such as the Deep Q-Network (DQN) have achieved state-of-the-art results in a variety of challenging, high-dimensional domains. This success is mainly attributed to the power of deep neural networks to learn rich domain representations for approximating the value function or policy. Batch reinforcement learning methods with linear representations, on the other hand, are more stable and require less hyper parameter tuning. Yet, substantial feature engineering is necessary to achieve good results. In this work we propose a hybrid approach – the Least Squares Deep Q-Network (LS-DQN), which combines rich feature representations learned by a DRL algorithm with the stability of a linear least squares method. We do this by periodically re-training the last hidden layer of a DRL network with a batch least squares update. Key to our approach is a Bayesian regularization term for the least squares update, which prevents over-fitting to the more recent data. We tested LS-DQN on five Atari games and demonstrate significant improvement over vanilla DQN and Double-DQN. We also investigated the reasons for the superior performance of our method. Interestingly, we found that the performance improvement can be attributed to the large batch size used by the LS method when optimizing the last layer.

## 1 Introduction

Reinforcement learning (RL) is a field of research that uses dynamic programing (DP; Bertsekas 2008), among other approaches, to solve sequential decision making problems. The main challenge in applying DP to real world problems is an exponential growth of computational requirements as the problem size increases, known as the curse of dimensionality (Bertsekas, 2008).

RL tackles the curse of dimensionality by *approximating* terms in the DP calculation such as the value function or policy. Popular function approximators for this task include deep neural networks, henceforth termed deep RL (DRL), and linear architectures, henceforth termed shallow RL (SRL).

SRL methods have enjoyed wide popularity over the years (see, e.g., Tsitsiklis et al. 1997; Bertsekas 2008 for extensive reviews). In particular, batch algorithms based on a least squares (LS) approach, such as Least Squares Temporal Difference (LSTD, Lagoudakis & Parr 2003) and Fitted-Q Iteration (FQI, Ernst et al. 2005) are known to be stable and data efficient. However, the success of these algorithms crucially depends on the quality of the feature representation. Ideally, the representation encodes rich, expressive features that can accurately represent the value function. However, in practice, finding such good features is difficult and often hampers the usage of linear function approximation methods.

In DRL, on the other hand, the features are learned *together with the value function* in a deep architecture. Recent advancements in DRL using convolutional neural networks demonstrated learning of expressive features (Zahavy et al., 2016; Wang et al., 2016) and state-of-the-art performance in challenging tasks such as video games (Mnih et al. 2015; Tessler et al. 2017; Mnih et al. 2016), and Go (Silver et al., 2016). To date, the most impressive DRL results (E.g., the works of Mnih et al. 2015, Mnih et al. 2016) were obtained using *online* RL algorithms, based on a stochastic gradient descent (SGD) procedure.

On the one hand, SRL is stable and data efficient. On the other hand, DRL learns powerful representations. This motivates us to ask: can we combine DRL with SRL to leverage the benefits of both?

In this work, we develop a *hybrid approach* that combines batch SRL algorithms with online DRL. Our main insight is that the last layer in a deep architecture can be seen as a linear representation, with the preceding layers encoding features. Therefore, the last layer can be learned using standard SRL algorithms. Following this insight, we propose a method that repeatedly *re-trains the last hidden layer* of a DRL network with a batch SRL algorithm, using data collected throughout the DRL run.

We focus on value-based DRL algorithms (e.g., the popular DQN of Mnih et al. 2015) and on SRL based on LS methods[1], and propose the Least Squares DQN algorithm (LS-DQN). Key to our approach is a novel regularization term for the least squares method that uses the DRL solution as a prior in a Bayesian least squares formulation. Our experiments demonstrate that this hybrid approach significantly improves performance on the Atari benchmark for several combinations of DRL and SRL methods.

To support our results, we performed an in-depth analysis to tease out the factors that make our hybrid approach outperform DRL. Interestingly, we found that the improved performance is mainly due to the large batch size of SRL methods compared to the small batch size that is typical for DRL.

## 2 Background

In this section we describe our RL framework and several shallow and deep RL algorithms that will be used throughout the paper.

**RL Framework:** We consider a standard RL formulation (Sutton & Barto, 1998) based on a Markov Decision Process (MDP). An MDP is a tuple $\langle S, A, R, P, \gamma \rangle$, where $S$ is a finite set of states, $A$ is a finite set of actions, and $\gamma \in [0, 1]$ is the discount factor. A transition probability function $P : S \times A \to \Delta_S$ maps states and actions to a probability distribution over next states. Finally, $R : S \times A \to [R_{min}, R_{max}]$ denotes the reward. The goal in RL is to learn a policy $\pi : S \to \Delta_A$ that solves the MDP by maximizing the expected discounted return $\mathbb{E}\left[\sum_{t=0}^{\infty} \gamma^t r_t \mid \pi\right]$. Value based RL methods make use of the action value function $Q^\pi(s, a) = \mathbb{E}[\sum_{t=0}^{\infty} \gamma^t r_t | s_t = s, a_t = a, \pi]$, which represents the expected discounted return of executing action $a \in A$ from state $s \in S$ and following the policy $\pi$ thereafter. The optimal action value function $Q^*(s, a)$ obeys a fundamental recursion known as the Bellman equation $Q^*(s, a) = \mathbb{E}\left[r_t + \gamma \max_{a'} Q^*(s_{t+1}, a') \mid s_t = s, a_t = a\right]$.

## 2.1 SRL algorithms

**Least Squares Temporal Difference Q-Learning (LSTD-Q):** LSTD (Barto & Crites, 1996) and LSTD-Q (Lagoudakis & Parr, 2003) are batch SRL algorithms. LSTD-Q learns a control policy $\pi$ from a batch of samples by estimating a linear approximation $\hat{Q}^\pi = \Phi w^\pi$ of the action value function $Q^\pi \in \mathbb{R}^{|S||A|}$, where $w^\pi \in \mathbb{R}^k$ are a set of weights and $\Phi \in \mathbb{R}^{|S||A| \times k}$ is a feature matrix. Each row of $\Phi$ represents a feature vector for a state-action pair $\langle s, a \rangle$. The weights $w^\pi$ are learned by enforcing $\hat{Q}^\pi$ to satisfy a fixed point equation w.r.t. the projected Bellman operator, resulting in a system of linear equations $Aw^\pi = b$, where $A = \Phi^T(\Phi - \gamma \mathbf{P}\Pi_\pi \Phi)$ and $b = \Phi^T \mathcal{R}$. Here, $\mathcal{R} \in \mathbb{R}^{|S||A|}$ is the reward vector, $\mathbf{P} \in \mathbb{R}^{|S||A| \times |S|}$ is the transition matrix and $\Pi_\pi \in \mathbb{R}^{|S| \times |S||A|}$ is a matrix describing the policy. Given a set of $N_{SRL}$ samples $D = \{s_i, a_i, r_i, s_{i+1}\}_{i=1}^{N_{SRL}}$, we can approximate $A$ and $b$ with the following empirical averages:

$$\tilde{A} = \frac{1}{N_{SRL}} \sum_{i=1}^{N_{SRL}} \left[ \phi(s_i, a_i)^T \Big( \phi(s_i, a_i) - \gamma \phi(s_{i+1}, \pi(s_{i+1})) \Big) \right], \quad \tilde{b} = \frac{1}{N_{SRL}} \sum_{i=1}^{N_{SRL}} \left[ \phi(s_i, a_i)^T r_i \right].$$

(1)

The weights $w^\pi$ can be calculated using a least squares minimization: $\tilde{w}^\pi = \arg\min_w \|\tilde{A}w - \tilde{b}\|_2^2$ or by calculating the pseudo-inverse: $\tilde{w}^\pi = \tilde{A}^\dagger \tilde{b}$. LSTD-Q is an *off-policy* algorithm: the same set of samples $D$ can be used to train any policy $\pi$ so long as $\pi(s_{i+1})$ is defined for every $s_{i+1}$ in the set.

**Fitted Q Iteration (FQI):** The FQI algorithm (Ernst et al., 2005) is a batch SRL algorithm that computes iterative approximations of the Q-function using regression. At iteration $N$ of the algorithm, the set $D$ defined above and the approximation from the previous iteration $Q^{N-1}$ are used to generate supervised learning targets: $y_i = r_i + \gamma \max_{a'} Q^{N-1}(s_{i+1}, a')$, $\forall i \in N_{SRL}$. These targets are then used by a supervised learning (regression) method to compute the next function in the sequence $Q^N$, by minimizing the MSE loss $Q^N = \arg\min_Q \sum_{i=1}^{N_{SRL}} (Q(s_i, a_i) - (r_i + \gamma \max_{a'} Q^{N-1}(s_{i+1}, a')))^2$. For a linear function approximation $Q_n(a, s) = \phi^T(s, a)w_n$, LS can be used to give the FQI solution $w_n = \arg\min_w \|\tilde{A}w - \tilde{b}\|_2^2$, where $\tilde{A}, \tilde{b}$ are given by:

$$\tilde{A} = \frac{1}{N_{SRL}} \sum_{i=1}^{N_{SRL}} \left[ \phi(s_i, a_i)^T \phi(s_i, a_i) \right], \qquad \tilde{b} = \frac{1}{N_{SRL}} \sum_{i=1}^{N_{SRL}} \left[ \phi(s_i, a_i)^T y_i \right] .$$

(2)

The FQI algorithm can also be used with non-linear function approximations such as trees (Ernst et al., 2005) and neural networks (Riedmiller, 2005). The DQN algorithm (Mnih et al., 2015) can be viewed as online form of FQI.

## 2.2 DRL algorithms

**Deep Q-Network (DQN):** The DQN algorithm (Mnih et al., 2015) learns the Q function by minimizing the mean squared error of the Bellman equation, defined as $\mathbb{E}_{s_t, a_t, r_t, s_{t+1}} \|Q_\theta(s_t, a_t) - y_t\|_2^2$, where $y_t = r_t + \gamma \max_{a'} Q_{\theta_{target}}(s_{t+1}, a')$. The DQN maintains two separate networks, namely the current network with weights $\theta$ and the target network with weights $\theta_{target}$. Fixing the target network makes the DQN algorithm equivalent to **FQI** (see the FQI MSE loss defined above), where the regression algorithm is chosen to be SGD (RMSPROP, Hinton et al. 2012). The DQN is an off-policy learning algorithm. Therefore, the tuples $\langle s_t, a_t, r_t, s_{t+1} \rangle$ that are used to optimize the network weights are first collected from the agent's experience and are stored in an Experience Replay (ER) buffer (Lin, 1993) providing improved stability and performance.

**Double DQN (DDQN):** DDQN (Van Hasselt et al., 2016) is a modification of the DQN algorithm that addresses overly optimistic estimates of the value function. This is achieved by performing action selection with the current network $\theta$ and evaluating the action with the target network, $\theta_{target}$, yielding the DDQN target update $y_t = r_t$ if $s_{t+1}$ is terminal, otherwise $y_t = r_t + \gamma Q_{\theta_{target}}(s_{t+1}, \max_a Q_\theta(s_{t+1}, a))$.

# 3 The LS-DQN Algorithm

We now present a hybrid approach for DRL with SRL updates[2]. Our algorithm, the LS-DQN Algorithm, periodically switches between training a DRL network and re-training its last hidden layer using an SRL method. [3]

We assume that the DRL algorithm uses a deep network for representing the Q function[4], where the last layer is linear and fully connected. Such networks have been used extensively in deep RL recently (e.g., Mnih et al. 2015; Van Hasselt et al. 2016; Mnih et al. 2016). In such a representation, the last layer, which approximates the Q function, can be seen as a linear combination of features (the output of the penultimate layer), and we propose to learn more accurate weights for it using SRL.

Explicitly, the LS-DQN algorithm begins by training the weights of a DRL network, $w_k$, using a value-based DRL algorithm for $N_{DRL}$ steps (Line 2). LS-DQN then updates the last hidden layer weights, $w_k^{last}$, by executing LS-UPDATE: retraining the weights using a SRL algorithm with $N_{SRL}$ samples (Line 3).

The LS-UPDATE consists of the following steps. First, data trajectories $D$ for the batch update are gathered using the current network weights, $w_k$ (Line 7). In practice, the current experience replay can be used and **no additional samples need to be collected**. The algorithm next generates new features $\Phi(s, a)$ from the data trajectories using the current DRL network with weights $w_k$. This step guarantees that we do not use samples with inconsistent features, as the ER contains features from 'old' networks weights. Computationally, this step requires running a forward pass of the deep network for every sample in $D$, and can be performed quickly using parallelization.

Once the new features are generated, LS-DQN uses an SRL algorithm to re-calculate the weights of the last hidden layer $w_k^{last}$ (Line 9).
While the LS-DQN algorithm is conceptually straightforward, we found that naively running it with off-the-shelf SRL algorithms such as FQI or LSTD resulted in instability and a degradation of the DRL performance. The reason is that the 'slow' SGD computation in DRL essentially retains information from older training epochs, while the batch SRL method 'forgets' all data but the most recent batch. In the following, we propose a novel regularization method for addressing this issue.

---

**Algorithm 1** LS-DQN Algorithm

---

**Require:**   $w_0$
1: **for** $k = 1 \cdots SRL_{iters}$ **do**
2:     $w_k \leftarrow$ trainDRLNetwork($w_{k-1}$)                    ▷ Train the DRL network for $N_{DRL}$ steps
3:     $w_k^{last} \leftarrow$ LS-UPDATE($w_k$)            ▷ Update the last layer weights with the SRL solution
4: **end for**
5:
6: **function** LS-UPDATE($w$)
7:     $D \leftarrow$ gatherData($w$)
8:     $\Phi(s, a) \leftarrow$ generateFeatures($D, w$)
9:     $w^{last} \leftarrow$ SRL-Algorithm($D, \Phi(s, a)$)
10:     **return** $w^{last}$
11: **end function**

---

**Regularization**

Our goal is to improve the performance of a value-based DRL agent using a batch SRL algorithm. Batch SRL algorithms, however, do not leverage the knowledge that the agent has gained before the most recent batch[5]. We observed that this issue prevents the use of off-the-shelf implementations of SRL methods in our hybrid LS-DQN algorithm.

To enjoy the benefits of both worlds, that is, a batch algorithm that can use the accumulated knowledge gained by the DRL network, we introduce a novel Bayesian regularization method for LSTD-Q and FQI that uses the last hidden layer weights of the DRL network $w_k^{last}$ as a *Bayesian prior* for the SRL algorithm [6].

**SRL Bayesian Prior Formulation:** We are interested in learning the weights of the last hidden layer ($w^{last}$), using a least squares SRL algorithm. We pursue a Bayesian approach, where the prior weights distribution at iteration $k$ of LS-DQN is given by $w_{prior} \sim N(w_k^{last}, \lambda^{-2})$, and we recall that $w_k^{last}$ are the last hidden layer weights of the DRL network at iteration $SRL_{iter} = k$. The Bayesian solution for the regression problem in the FQI algorithm is given by (Box & Tiao, 2011)

$$w^{last} = (\tilde{A} + \lambda I)^{-1}(\tilde{b} + \lambda w_k^{last}) \ ,$$

where $\tilde{A}$ and $\tilde{b}$ are given in Equation 2. A similar regularization can be added to LSTD-Q based on a regularized fixed point equation (Kolter & Ng, 2009). Full details are in Appendix A.

# 4 Experiments

In this section, we present experiments showcasing the improved performance attained by our LS-DQN algorithm compared to state-of-the-art DRL methods. Our experiments are divided into three sections. In Section 4.1, we start by investigating the behavior of SRL algorithms in high dimensional environments. We then show results for the LS-DQN on five Atari domains, in Section 4.2, and compare the resulting performance to regular DQN and DDQN agents. Finally, in Section 4.3, we present an ablative analysis of the LS-DQN algorithm, which clarifies the reasons behind our algorithm's success.

## 4.1 SRL Algorithms with High Dimensional Observations

In the first set of experiments, we explore how least squares SRL algorithms perform in domains with high dimensional observations. This is an important step before applying a SRL method within the LS-DQN algorithm. In particular, we focused on answering the following questions: (1) What regularization method to use? (2) How to generate data for the LS algorithm? (3) How many policy improvement iterations to perform?

To answer these questions, we performed the following procedure: We trained DQN agents on two games from the Arcade Learning Environment (ALE, Bellemare et al.); namely, Breakout and Qbert, using the vanilla DQN implementation (Mnih et al., 2015). For each DQN run, we (1) **periodically** [7] save the current DQN network weights and ER; (2) Use an SRL algorithm (LSTD-Q or FQI) to re-learn the weights of the last layer, and (3) evaluate the resulting DQN network by temporarily replacing the DQN weights with the SRL solution weights. After the evaluation, **we replace back the original DQN weights** and continue training.

Each evaluation entails 20 roll-outs [8] with an $\epsilon$-greedy policy (similar to Mnih et al., $\epsilon = 0.05$). This periodic evaluation setup allowed us to effectively experiment with the SRL algorithms and obtain clear comparisons with DQN, without waiting for full DQN runs to complete.

**(1) Regularization:** Experiments with standard SRL methods without any regularization yielded poor results. We found the main reason to be that the matrices used in the SRL solutions (Equations 1 and 2) are ill-conditioned, resulting in instability. One possible explanation stems from the sparseness of the features. The DQN uses ReLU activations (Jarrett et al., 2009), which causes the network to learn sparse feature representations. For example, once the DQN completed training on Breakout, 96% of features were zero.

Once we added a regularization term, we found that the performance of the SRL algorithms improved. We experimented with the $\ell_2$ and Bayesian Prior (BP) regularizers ($\lambda \in [0, 10^2]$). While the $\ell_2$ regularizer showed competitive performance in Breakout, we found that the BP performed better across domains (Figure 1, best regularizers chosen, shows the average score of each configuration following the explained evaluation procedure, for the different epochs). Moreover, the BP regularizer

was not sensitive to the scale of the regularization coefficient. Regularizers in the range $(10^{-1}, 10^{1})$ performed well across all domains. A table of average scores for different coefficients can be found in Appendix C.1. Note that we do not expect for much improvement as we replace back the original DQN weights after evaluation.

**(2) Data Gathering:** We experimented with two mechanisms for generating data: (1) generating new data from the current policy, and (2) using the ER. We found that the data generation mechanism had a significant impact on the performance of the algorithms. When the data is generated only from the current DQN policy (without ER) the SRL solution resulted in poor performance compared to a solution using the ER (as was observed by Mnih et al. 2015). We believe that the main reason the ER works well is that the ER contains data sampled from multiple (past) policies, and therefore exhibits more exploration of the state space.

**(3) Policy Improvement:** LSTD-Q and FQI are off-policy algorithms and can be applied iteratively on the same dataset (e.g. LSPI, Lagoudakis & Parr 2003). However, in practice, we found that performing multiple iterations did not improve the results. A possible explanation is that by improving the policy, the policy reaches new areas in the state space that are not represented well in the current ER, and therefore are not approximated well by the SRL solution and the current DRL network.

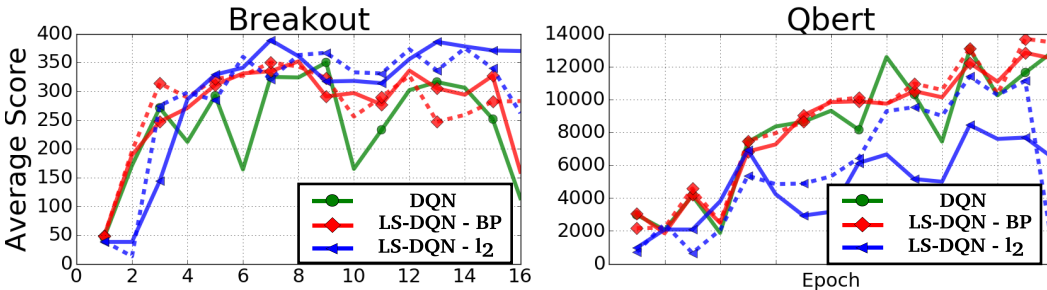

Figure 1: Periodic evaluation for DQN (green), **LS-DQN$_{\text{LSTD-Q}}$** with Bayesian prior regularization (red, solid $\lambda = 10$, dashed $\lambda = 1$), and $\ell_2$ regularization (blue, solid $\lambda = 0.001$, dashed $\lambda = 0.0001$).

## 4.2 Atari Experiments

We next ran the full LS-DQN algorithm (Alg. 1) on five Atari domains: Asterix, Space Invaders, Breakout, Q-Bert and Bowling. We ran the LS-DQN using both DQN and DDQN as the DRL algorithm, and using both LSTD-Q and FQI as the SRL algorithms. We chose to run a LS-update every $N_{DRL} = 500k$ steps, for a total of 50M steps ($SRL_{iters} = 100$). We used the current ER buffer as the 'generated' data in the LS-UPDATE function (line 7 in Alg. 1, $N_{SRL} = 1M$), and a regularization coefficient $\lambda = 1$ for the Bayesian prior solution (both for FQI and LSTQ-Q). We emphasize the we did not use any additional samples beyond the samples already obtained by the DRL algorithm.

Figure 2 presents the learning curves of the DQN network, LS-DQN with LSTD-Q, and LS-DQN with FQI (referred to as **DQN**, **LS-DQN$_{\text{LSTD-Q}}$**, and **LS-DQN$_{\text{FQI}}$**, respectively) on three domains: Asterix, Space Invaders and Breakout. Note that we use the same evaluation process as described in Mnih et al. (2015). We were also interested in a test to measure differences between learning curves, and not only their maximal score. Hence we chose to perform Wilcoxon signed-rank test on the average scores between the three DQN variants. This non-parametric statistical test measures whether related samples differ in their means (Wilcoxon, 1945). We found that the learning curves for both LS-DQN$_{\text{LSTD-Q}}$ and LS-DQN$_{\text{FQI}}$ were statistically significantly better than those of DQN, with p-values smaller than 1e-15 for all three domains.

Table 1 presents the maximum average scores along the learning curves of the five domains, when the SRL algorithms were incorporated into both DQN agents and DDQN agents (the notation is similar, i.e., **LS-DDQN$_{\text{FQI}}$**)[9]. Our algorithm, LS-DQN, attained better performance compared to the

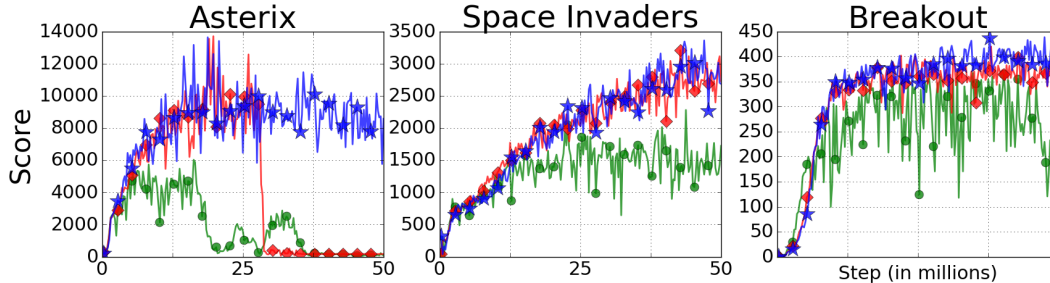

Figure 2: Learning curves of DQN (green), LS-DQN$_{\text{LSTD-Q}}$ (red), and LS-DQN$_{\text{FQI}}$ (blue).

vanilla DQN agents, as seen by the higher scores in Table 1 and Figure 2. We observe an interesting phenomenon for the game Asterix: In Figure 2, the DQN's score "crashes" to zero (as was observed by Van Hasselt et al. 2016). LS-DQN$_{\text{LSTD-Q}}$ did not manage to resolve this issue, even though it achieved a significantly higher score that that of the DQN. LS-DQN$_{\text{FQI}}$, however, maintained steady performance and did not "crash" to zero. We found that, in general, incorporating FQI as an SRL algorithm into the DRL agents resulted in improved performance.

Table 1: Maximal average scores across five different Atari domains for each of the DQN variants.

| Algorithm \ Game | Breakout | Space Invaders | Asterix | Qbert | Bowling |
|---|---|---|---|---|---|
| DQN[9] | 401.20 | 1975.50 | 6011.67 | 10595.83 | 42.40 |
| LS-DQN$_{\text{LSTD-Q}}$ | 420.00 | 3207.44 | **13704.23** | 10767.47 | 61.21 |
| LS-DQN$_{\text{FQI}}$ | **438.55** | **3360.81** | 13636.81 | **12981.42** | **75.38** |
| DDQN[9] | 375.00 | 3154.60 | 15150.00 | **14875.00** | 70.50 |
| LS-DDQN$_{\text{FQI}}$ | **397.94** | **4400.83** | **16270.45** | 12727.94 | **80.75** |

## 4.3 Ablative Analysis

In the previous section, we saw that the LS-DQN algorithm has improved performance, compared to the DQN agents, across a number of domains. The goal of this section is to understand the reasons behind the LS-DQN's improved performance by conducting an ablative analysis of our algorithm. For this analysis, we used a DQN agent that was trained on the game of Breakout, in the same manner as described in Section 4.1. We focus on analyzing the **LS-DQN$_{\text{FQI}}$** algorithm, that has the same optimization objective as DQN (cf. Section 2), and postulate the following conjectures for its improved performance:

(i) The SRL algorithms use a Bayesian regularization term, which is not included in the DQN objective.

(ii) The SRL algorithms have less hyperparameters to tune and generate an explicit solution compared to SGD-based DRL solutions.

(iii) Large-batch methods perform better than small-batch methods when combining DRL with SRL.

(iv) SRL algorithms focus on training the last layer and are easier to optimize.

**The Experiments:** We started by analyzing the learning method of the last layer (i.e., the 'shallow' part of the learning process). We did this by optimizing the last layer, at each LS-UPDATE epoch, using (1) FQI with a Bayesian prior and a LS solution, and (2) an ADAM (Kingma & Ba, 2014) optimizer with and without an additional Bayesian prior regularization term in the loss function. We compared these approaches for different mini-batch sizes of 32, 512, and 4096 data points, and used $\lambda = 1$ for all experiments.

Relating to conjecture (ii), note that the FQI algorithm has only one hyper-parameter to tune and produces an explicit solution using the whole dataset simultaneously. ADAM, on the other hand, has more hyper-parameters to tune and works on different mini-batch sizes.

**The Experimental Setup:** The experiments were done in a periodic fashion similar to Section 4.1, i.e., testing behavior in different epochs over a vanilla DQN run. For both ADAM and FQI, we first collected $80k$ data samples from the ER at each epoch. For ADAM, we performed 20 iterations over the data, where each iteration consisted of randomly permuting the data, dividing it into mini-batches and optimizing using ADAM over the mini-batches[10]. We then simulate the agent and report average scores across 20 trajectories.

**The Results:** Figure 3 depicts the difference between the average scores of (1) and (2) to that of the DQN baseline scores. We see that larger mini-batches result in improved performance. Moreover, the LS solution (FQI) outperforms the ADAM solutions for mini-batch sizes of 32 and 512 on most epochs, and even slightly outperforms the best of them (mini-batch size of 4096 and a Bayesian prior). In addition, a solution with a prior performs better than a solution without a prior.

**Summary:** Our ablative analysis experiments strongly support conjectures (iii) and (iv) from above, for explaining LS-DQN's improved performance. That is, large-batch methods perform better than small-batch methods when combining DRL with SRL as explained above; and SRL algorithms that focus on training only the last layer are easier to optimize, as we see that optimizing the last layer improved the score across epochs.

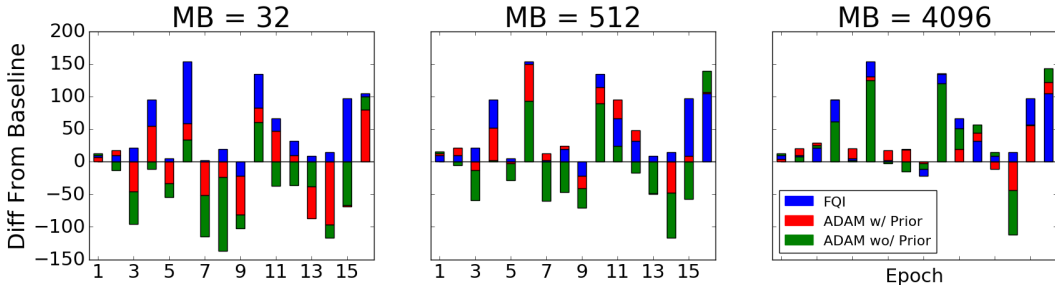

Figure 3: Differences of the average scores, for different learning methods, compared to vanilla DQN. Positive values represent improvement over vanilla DQN. For example, for mini-batch of 32 (left figure), in epoch 3, FQI (blue) out-performed vanilla DQN by 21, while ADAM with prior (red), and ADAM without prior (green) under-performed by -46, and -96, respectively. Note that: (1) as the mini-batch size increases, the improvement of ADAM over DQN becomes closer to the improvement of FQI over DQN, and (2) optimizing the last layer improves performance.

We finish this Section with an interesting observation. While the LS solution improves the performance of the DRL agents, we found that the LS solution weights are very close to the baseline DQN solution. See Appendix D, for the full results. Moreover, the distance was inversely proportional to the performance of the solution. That is, the FQI solution that performed the best, was the closest (in $\ell_2$ norm) to the DQN solution, and vice versa. There were orders of magnitude differences between the norms of solutions that performed well and those that did not. Similar results, i.e., that large-batch solutions find solutions that are close to the baseline, have been reported in (Keskar et al., 2016). We further compare our results with the findings of Keskar et al. in the section to follow.

# 5 Related work

We now review recent works that are related to this paper.

**Regularization:** The general idea of applying regularization for feature selection, and to avoid overfitting is a common theme in machine learning. However, applying it to RL algorithms is challenging due to the fact that these algorithms are based on finding a fixed-point rather than optimizing a loss function (Kolter & Ng, 2009).Value-based DRL approaches do not use regularization layers (e.g. pooling, dropout and batch normalization), which are popular in other deep learning methods. The DQN, for example, has a relatively shallow architecture (three convolutional layers, followed by two fully connected layers) without any regularization layers. Recently, regularization was introduced

in problems that combine value-based RL with other learning objectives. For example, Hester et al. (2017) combine RL with supervised learning from expert demonstration, and introduce regularization to avoid over-fitting the expert data; and Kirkpatrick et al. (2017) introduces regularization to avoid catastrophic forgetting in transfer learning. SRL methods, on the other hand, perform well with regularization (Kolter & Ng, 2009) and have been shown to converge Farahmand et al. (2009).

**Batch size:** Our results suggest that a large batch LS solution for the last layer of a value-based DRL network can significantly improve it's performance. This result is somewhat surprising, as it has been observed by practitioners that using larger batches in deep learning degrades the quality of the model, as measured by its ability to generalize (Keskar et al., 2016).

However, our method differs from the experiments performed by Keskar et al. 2016 and therefore does not contradict them, for the following reasons: (1) The LS-DQN Algorithm uses the large batch solution only for the last layer. The lower layers of the network are not affected by the large batch solution and therefore do not converge to a sharp minimum. (2) The experiments of (Keskar et al., 2016) were performed for classification tasks, whereas our algorithm is minimizing an MSE loss. (3) Keskar et al. showed that large-batch solutions work well when piggy-backing (warm-started) on a small-batch solution. Similarly, our algorithm mixes small and large batch solutions as it switches between them periodically.

Moreover, it was recently observed that flat minima in practical deep learning model classes can be turned into sharp minima via re-parameterization without changing the generalization gap, and hence it requires further investigation Dinh et al. (2017). In addition, Hoffer et al. showed that large-batch training can generalize as well as small-batch training by adapting the number of iterations Hoffer et al. (2017). Thus, we strongly believe that our findings on combining large and small batches in DRL are in agreement with recent results of other deep learning research groups.

**Deep and Shallow RL:** Using the last-hidden layer of a DNN as a feature extractor and learning the last layer with a different algorithm has been addressed before in the literature, e.g., in the context of transfer learning (Donahue et al., 2013). In RL, there have been competitive attempts to use SRL with unsupervised features to play Atari (Liang et al., 2016; Blundell et al., 2016), and to learn features automatically followed by a linear control rule (Song et al., 2016), but to the best of our knowledge, this is the first attempt that successfully combines DRL with SRL algorithms.

# 6   Conclusion

In this work we presented LS-DQN, a hybrid approach that combines least-squares RL updates within online deep RL. LS-DQN obtains the best of both worlds: rich representations from deep RL networks as well as stability and data efficiency of least squares methods. Experiments with two deep RL methods and two least squares methods revealed that a hybrid approach consistently improves over vanilla deep RL in the Atari domain. Our ablative analysis indicates that the success of the LS-DQN algorithm is due to the large batch updates made possible by using least squares.

This work focused on value-based RL. However, our hybrid linear/deep approach can be extended to other RL methods, such as actor critic (Mnih et al., 2016). More broadly, decades of research on linear RL methods have provided methods with strong guarantees, such as approximate linear programming (Desai et al., 2012) and modified policy iteration (Scherrer et al., 2015). Our approach shows that with the correct modifications, such as our Bayesian regularization term, linear methods can be combined with deep RL. This opens the door to future combinations of well-understood linear RL with deep representation learning.

**Acknowledgement**   This research was supported by the European Community's Seventh Framework Program (FP7/2007-2013) under grant agreement 306638 (SUPREL). A. Tamar is supported in part by Siemens and the Viterbi Scholarship, Technion.

## Footnotes

[1] Our approach can be generalized to other DRL/SRL variants.

[2]Code is available online at `https://github.com/Shallow-Updates-for-Deep-RL`

[3]We refer the reader to Appendix B for a diagram of the algorithm.

[4]The features in the last DQN layer are not action dependent. We generate action-dependent features $\Phi(s, a)$ by zero-padding to a one-hot state-action feature vector. See Appendix E for more details.

[5]While conceptually, the data batch can include *all* the data seen so far, due to computational limitations, this is not a practical solution in the domains we consider.

[6]The reader is referred to Ghavamzadeh et al. (2015) for an overview on using Bayesian methods in RL.

[7]Every three million DQN steps, referred to as one epoch (out of a total of 50 million steps).

[8]Each roll-out starts from a new (random) game and follows a policy until the agent loses all of its lives.

[9] Scores for DQN and DDQN were taken from Van Hasselt et al. (2016).

[10] The selected hyper-parameters used for these experiments can be found in Appendix D, along with results for one iteration of ADAM.

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
