[Supplementary Material · shallow_updates_supp.pdf]

# A    Adding Regularization to LSTD-Q

For LSTD-Q, regularization cannot be applied directly since the algorithm is finding a fixed-point and not solving a LS problem. To overcome this obstacle, we augment the fixed point function of the LSTD-Q algorithm to include a regularization term based on (Kolter & Ng, 2009):

$$f(w) = \underset{u}{\mathrm{argmin}} \, \|\phi u - \Pi T^* \phi w\| + \lambda g(u) \ , \tag{3}$$

where $\Pi$ stands for the linear projection, $T^*$ for the Bellman optimality operator and $g(u)$ is the regularization function. Once the augmented problem is solved, the solution to the regularized LSTD-Q problem is given by $w = f(w)$. This derivation results in the same solution for LSTD-Q as was obtained for FQI (Equation 2). In the special case where $\mu = 0$, we get the $L_2$ regularized solution of Kolter & Ng (2009).

# B    LS-DQN Algorithm

Figure 4 provides an overview of the LS-DQN algorithm described in the main paper. The DNN agent is trained for $N_{DRL}$ steps (A). The weights of the last hidden layer are denoted $w_k$. Data is then gathered (LS.1) from the agent's experience replay and features are generated (LS.2). An SRL-Algorithm is applied to the generated features (LS.3) which includes a regularized Bayesian prior weight update (LS.4). Note that the weights $w_k$ are used as the prior. The weights of the last hidden layer are then replaced by the SRL output $w^{last}$ and this process is repeated.

Figure 4: An overview of the LS-DQN algorithm.

## C   Results for SRL Algorithms with High Dimensional Observations

We present the average scores (averaged over 20 roll-outs) at different epochs, for both the original DQN and after relearning the last layer using LSTD-Q, for different regularization coefficients.

**Breakout**

Table 2: Average scores on the different epochs as a function of regularization coefficients

| Epoch \ $\lambda$ | $10^2$ | $10^1$ | $10^0$ | $10^{-1}$ | $10^{-2}$ | $10^{-3}$ | $10^{-4}$ | $10^{-5}$ | $10^{-6}$ | $10^{-7}$ | DQN |
|---|---|---|---|---|---|---|---|---|---|---|---|
| Epoch 1 | 54 | 49 | 48 | 44 | 53 | 49 | 48 | 50 | 28 | 30 | 46 |
| Epoch 2 | 207 | 189 | 196 | 193 | 64 | 30 | 18 | 4 | 9 | 5 | 171 |
| Epoch 3 | 238 | 247 | 314 | 284 | 277 | 254 | 270 | 232 | 225 | 194 | 271 |
| Epoch 4 | 238 | 271 | 289 | 249 | 207 | 201 | 291 | 326 | 274 | 304 | 212 |
| Epoch 5 | 265 | 311 | 322 | 315 | 208 | 109 | 175 | 36 | 14 | 48 | 292 |
| Epoch 6 | 299 | 331 | 327 | 328 | 259 | 150 | 248 | 227 | 281 | 245 | 164 |
| Epoch 7 | 332 | 335 | 350 | 266 | 128 | 67 | 145 | 249 | 291 | 214 | 325 |
| Epoch 8 | 361 | 352 | 343 | 262 | 204 | 65 | 270 | 309 | 287 | 304 | 324 |
| Epoch 9 | 294 | 291 | 323 | 319 | 101 | 85 | 224 | 276 | 347 | 340 | 350 |
| Epoch 10 | 186 | 297 | 256 | 263 | 243 | 236 | 349 | 323 | 333 | 333 | 165 |
| Epoch 11 | 241 | 277 | 290 | 140 | 79 | 111 | 338 | 335 | 330 | 315 | 233 |
| Epoch 12 | 328 | 336 | 327 | 352 | 226 | 208 | 337 | 374 | 354 | 377 | 302 |
| Epoch 13 | 343 | 305 | 247 | 308 | 62 | 112 | 338 | 342 | 305 | 344 | 316 |
| Epoch 14 | 278 | 294 | 259 | 273 | 156 | 198 | 320 | 355 | 350 | 346 | 306 |
| Epoch 15 | 312 | 327 | 282 | 292 | 161 | 141 | 321 | 381 | 368 | 367 | 252 |
| Epoch 16 | 186 | 160 | 283 | 273 | 170 | 225 | 370 | 314 | 325 | 324 | 114 |

**Qbert**

Table 3: Average scores on the different epochs as a function of regularization coefficients

| Epoch \ $\lambda$ | $10^2$ | $10^1$ | $10^0$ | $10^{-1}$ | $10^{-2}$ | $10^{-3}$ | $10^{-4}$ | $10^{-5}$ | $10^{-6}$ | $10^{-7}$ | DQN |
|---|---|---|---|---|---|---|---|---|---|---|---|
| Epoch 1 | 3470 | 3070 | 2163 | 1998 | 1599 | 2078 | 964 | 629 | 831 | 484 | 2978 |
| Epoch 2 | 2794 | 1853 | 2196 | 2565 | 3839 | 3558 | 1376 | 2123 | 1728 | 2388 | 2060 |
| Epoch 3 | 4253 | 4188 | 4579 | 4034 | 4031 | 2239 | 561 | 691 | 824 | 570 | 4148 |
| Epoch 4 | 2789 | 2489 | 2536 | 2750 | 3435 | 5214 | 2730 | 2303 | 1356 | 594 | 1878 |
| Epoch 5 | 6426 | 6831 | 7480 | 6703 | 3419 | 3335 | 4205 | 3519 | 4673 | 5231 | 7410 |
| Epoch 6 | 8480 | 7265 | 7950 | 5300 | 4978 | 4178 | 4533 | 6005 | 6133 | 4829 | 8356 |
| Epoch 7 | 8176 | 9036 | 8635 | 7774 | 7269 | 7428 | 6196 | 3030 | 3246 | 2343 | 8643 |
| Epoch 8 | 9104 | 10340 | 9935 | 7293 | 7689 | 7343 | 6728 | 2913 | 3299 | 1473 | 9315 |
| Epoch 9 | 9274 | 10288 | 9115 | 7508 | 6660 | 7800 | 120 | 8133 | 4880 | 5018 | 8156 |
| Epoch 10 | 10523 | 7245 | 9704 | 7949 | 8640 | 7794 | 2663 | 8905 | 10044 | 7585 | 12584 |
| Epoch 11 | 10821 | 11510 | 9971 | 7064 | 6836 | 9908 | 1020 | 11868 | 9940 | 11138 | 10290 |
| Epoch 12 | 7291 | 10134 | 7583 | 6673 | 7815 | 9028 | 5564 | 8893 | 8649 | 6748 | 7438 |
| Epoch 13 | 12365 | 12220 | 13103 | 11868 | 11531 | 10091 | 2753 | 10804 | 8216 | 8835 | 13054 |
| Epoch 14 | 11686 | 11085 | 10338 | 10811 | 8386 | 9580 | 2980 | 6469 | 6435 | 6071 | 10249 |
| Epoch 15 | 11228 | 12841 | 13696 | 10971 | 5820 | 10148 | 7524 | 11959 | 9270 | 6949 | 11630 |
| Epoch 16 | 11643 | 12489 | 13468 | 11773 | 8191 | 8976 | 198 | 7284 | 7598 | 5649 | 12923 |

## D   Results for Ablative Analysis

We used the implementation of ADAM from the `optim` package for torch that can be found at `https://github.com/torch/optim/blob/master/adam.lua`. We used the default hyperparameters (except for the learning rate): learningRate= 0.00025, learningRateDecay= 0, beta1= 0.9, beta2= 0.999, epsilon= 1e−8, and weightDecay= 0. For solutions that use the prior, we set $\lambda = 1$.

Figure 5 depicts the offset of the average scores from the DQN's scores, after one iteration of the ADAM algorithm:

Figure 5: Differences of the average scores from DQN compared to ADAM and FQI (with and without priors) for different mini-batches (MB) sizes.

Table 4 shows the norm of the difference between the different solution weights and the original last layer weights of the DQN (divided by the norm of the DQN's weights for scale), averaged over epochs. Note that MB stands for mini-batch sizes used by the ADAM solver.

Table 4: Norms of the Difference Between solutions Weights

|  | Batch | MB=32 iter=1 | MB=32 iter=20 | MB=512 iter=1 | MB=512 iter=20 | MB=4096 iter=1 | MB=4096 iter=20 |
|---|---|---|---|---|---|---|---|
| w/ prior | ~3e-4 | ~3e-3 | ~3e-3 | ~2e-3 | ~2e-3 | ~1.7e-3 | ~1.8e-3 |
| wo/ prior |  | ~3.8e-2 | ~2.7e-1 | ~1.3e-2 | ~1.2e-1 | ~5e-3 | ~5e-2 |

# E  Feature augmentation

The LS-DQN algorithm requires a function $\Phi(s, a)$ that creates features (Algorithm 1, Line 9) for a dataset $D$ using the current value-based DRL network. Notice that for most value-based DRL networks (e.g. DQN and DDQN), the DRL features (output of the last hidden layer) are a function of the state and not a function of the action. On the other hand, the FQI and LSTDQ algorithms require features that are a function of both state and action. We, therefore, augment the DRL features to be a function of the action in the following manner. Denote by $\phi(s) \in \mathbb{R}^f$ the output of the last hidden layer in the DRL network (where $f$ is the number of neurons in this layer). We define $\Phi(s, a) \in \mathbb{R}^{f|A|}$ to be $\phi(s)$ on a subset of indices that belongs to action $a$ and zero otherwise, where $|A|$ refers to the size of the action space.

Note that in practice, DQN and DDQN maintain an ER, and we create features for all the states in the ER. A more computationally efficient approach would be to store the features in the ER after the DRL agent visits them, makes a forward propagation (and compute features) and store them in the ER. However, SRL algorithms work only with features that are fixed over time. Therefore, we generate new features with the current DRL network.