[Reviews · NeurIPS 2017]

Reviewer 1



The paper proposes an add on to DQN algorithm, by using an additional "periodical re-training the last hidden layer of a DRL network with a batch least squares update". The authors explain the logic behind the idea really well, supported by the context of Deep RL and SRL algorithms. The paper first analyzes the behavior of SRL algorithms with high dimensional problems, then they compare the proposed algorithm in 3 different atari domains. The results show mostly improvement over DQN and DDQN algorithms on maximum results. The proposed algorithm has a more stable learning curve compared to DQN. In my opinion, the motivation is somewhat comparable to DDQN, and address the same weakness of the DQN algorithm. On the other hand being able to combine the shallow updates with even DDQN without any additional samples, is a definite plus. In the results section, the authors present the maximal average scores reached for different algorithms, but I was curious about the confidence intervals of these, and how do they take the average of maximum? Is it average over a sliding window, or multiple runs? I liked the idea behind the Section 4.3, but I find it hard to follow / understand. Especially the figure 4, could be represented in a different way instead of incremental bar charts. I would replace the sentence "We discuss various works related to our approach." with something more informative, or just remove it. Overall I think the paper brings a good research direction with interesting results and discssion.

Reviewer 2



This paper describes a relatively straightforward RL algorithm that involves occasionally re-training the last layer of a DQN net with a batch-based linear RL algorithm. The authors hope that the batch-based approach will converge to a more precise value-function for the given step of the overall algorithm, thus improving the policy's performance scores at that step. By occasionally re-fitting the acting policy's value function more precisely through this batch step, the final overall learning algorithm will be less hindered by temporary artefacts due to the slowness of SGD, and therefore may hopefully converge both more stably and to a better final policy. Opinion: The paper is well-written and easy to follow. The experiments are well-structured and equally easy to understand. I would have appreciated a little more detail as to the Bayesian prior's nuances, esp. relatively to L2 with which I am much more familiar. What makes this particular prior so capable in maintaining stability during an LS pass? The experiments seem to defend the claims in the paper, but I am having a bit of trouble with the final paragraph's suggestion that the best performing solutions were those that were closest to the original weights in L2. Does this suggest that the LS step is simply 'cleaning' up a couple artefacts that the SGD has not quite brought to convergence? More in-depth analysis of this point would be interesting, as I'm not entirely sure I understand /what/ exactly makes this work much better, other than a more robust value-iteration step that avoids bizarreness that can be present in an SGD-converged solution... Also, please clean up your plots with reduced line thickness and alpha transparency on your lines, and also review your font sizes. These plots are pretty ugly and a minimum amount of effort could make them just fine and easier t read ('import seaborn' should get you 90% of the way there).

Reviewer 3



The authors propose to augment value-based methods for deep reinforcement learning (DRL) with batch methods for linear approximation function (SRL). The idea is motivated by interpreting the output of the second-to-last layer of a neural network as linear features. In order to make this combination work, the authors argue that regularization is needed. Experimental results are provided for 5 Atari games on combinations of DQN/Double DQN and LSTD-Q/FQI. Strengths: I find the proposition of combining DRL and SRL with Bayesian regularization original and promising. The explanation provided for the improved performance seems reasonable, but it could have been better validated (see below). Weaknesses: The presentation of Algorithm 1 and in particular line 7, is a bit strange to me, given that in Section 4, the authors mention that generating D with the current weights results in poor performance. Why not present line 7 as using ER only? Besides, would it be interesting to use p% of trajectories from ER and (1-p)% of trajectories generated from the current weights? The experiments could be more complete. For instance, in Table 1, the results for LS-DDQN_{LSTD-Q} are missing and in Fig.2, the curves for Qbert and Bowling are not provided. For a fairer comparison between DQN and LS-DQN, I think the authors should compare Algorithm 1 against DQN given an equal time-budget to check that their algorithm indeed provides at the end a better performance. Besides, the ablative analysis was just performed on Breakout. Would the same conclusions hold on different games? Detailed comments: l.41: the comment on data-efficiency of SRL is only valid in the batch setting, isn’t it? l.114, l.235: it’s l.148: in the equation above, \mu should be w_k. I find the notation w_k for the last hidden layer to be unsatisfying, as w_k also denotes the weights of the whole network at iteration k l.205, l.245: a LS -> an LS l.222: algortihm